# FVM-RANS Modeling of Air Pollutants Dispersion and Traffic Emission in Dhaka City on a Suburb Scale

Md. Eabad Ali [1], Md Farhad Hasan [2,3], Sadia Siddiqa [4], Md. Mamun Molla [5,6], Most. Nasrin Akhter [1,*]

1   Department of Mathematics, Dhaka University of Engineering & Technology, Gazipur 1707, Bangladesh
2   Victoria State Government, Melbourne, VIC 3083, Australia
3   School of Computing, Engineering and Mathematical Sciences, La Trobe University, Melbourne, VIC 3086, Australia
4   Department of Mathematics & Sciences, College of Humanities & Sciences, Prince Sultan University, Riyadh 11586, Saudi Arabia
5   Department of Mathematics & Physics, North South University, Dhaka 1229, Bangladesh
6   Center for Applied Scientific Computing (CASC), North South University, Dhaka 1229, Bangladesh
*   Correspondence: nasrin@duet.ac.bd; Tel.: +880-1749288044

**Abstract:** The present study aims to investigate the impact of air pollutants dispersion from traffic emission under the influence of wind velocity and direction considering the seasonal cycle in two major areas of Dhaka city: namely, Tejgaon and Gazipur. Carbon monoxide (CO) mass fraction has been considered as a representative element of traffic-exhausted pollutants, and the distribution of pollutants has been investigated in five different street geometries: namely, single regular and irregular, double regular and irregular, and finally, multiple irregular streets. After the grid independence test confirmation as well as numerical validation, a series of case studies has been presented to analyze the air pollutants dispersion, which mostly exists due to the traffic emission. The popular Reynolds-averaged Navier–Stokes (RANS) approach has been considered, and the finite volume method (FVM) has been applied by ANSYS Fluent$^{TM}$. The $k - \epsilon$ turbulence model has been integrated from the RANS approach. It was found that the wind velocity as well as wind direction and the fluid flow fields can play a potential role on pollution dispersion in the Dhaka city street canyons and suburbs. Inhabitants residing near the single regular streets are exposed to more traffic emission than those of single irregular streets due to fewer obstacles being created by the buildings. Double regular streets have been found to be a better solution to disperse pollutants, but city dwellers in the east region of double irregular streets are exposed to a greater concentration of pollutants due to the change of wind directions and seasonal cycles. Multiple irregular streets limit the mobility of the pollutants due to the increased number of buildings, yet the inhabitants near the multi-irregular streets are likely to experience approximately 11.25% more pollutants than other dwellers living far from the main street. The key findings of this study will provide insights on improving the urbanization plan where different geometries of streets are present and city dwellers could have less exposure to traffic-exhausted pollutants. The case studies will also provide a template layout to map pollutant exposure to identify the alarming zone and stop incessant building construction within those regions by creating real-time air quality monitoring to safeguard public safety.

**Keywords:** pollutants dispersion; street canyons; RANS; traffic emission; finite volume method; environmental aerodynamics

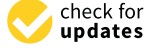



## 1. Introduction

Over the last few decades, incessant pollutant dispersion within urban areas has been significantly downgrading the living conditions. As a result, the number of detrimental issues is rapidly increasing at an alarming rate, affecting the environmental sustainability [1–3]. With the rapid blooming of urbanization, the majority of the people across the

world is living in urban areas. According to a report by United Nations, there were approximately 751 million urban inhabitants in 1950, which increased to 4.2 billion in 2018 [4]. It is also statistically anticipated that the number will reach approximately 6.4 billion by 2050 [4]. Urbanization is crucial for economic development, particularly ever since the COVID-19 pandemic, another major setback in public health, started to adversely affect the global economy. In fact, over 80% of the urban inhabitants are found to be inhaling polluted air over the World Health Organization (WHO) limits [5]. In addition to the COVID-19 pandemic, approximately 4.2 million deaths across the globe have been reported due to the air pollution [5]. During the COVID-19 global pandemic, the environmental pollutants nitrogen dioxide ($NO_2$) and carbon monoxide (CO) were drastically reduced, and the industrial activities were also in short-range with the spreading of this disease. As a result, the concentration of the $NO_2$ and CO pollutants in 2020 was 5% lower than that in 2019 [6]. Considering the significant growth in urban population and the involuntary vulnerability of inhabitants to the urban pollution, the WHO and other concerned authorities have already declared air pollution as a serious global public health issue [4,5]. Therefore, attaining the sustainable development milestones will remain uncertain unless public health is not improved and proper understanding on the sources and solutions to the air pollution is not comprehended.

The primary causes to the growing urban air pollution are related to traffic, engine combustion, fuel consumption in different set-ups, and industrial activities [7,8]. The air pollution due to traffic has been attributed to leading this unpleasant list as being the source of almost 25% of the total urban air pollution [7,8]. The coupled problem from the traffic emission and the stagnated airflow as a consequence of densely packed high-rise buildings create further complications for the inhabitants. Improper and unplanned ventilation systems further reduce the indoor air quality and since the stagnated outdoor pollutants can access the indoor in high-rise structures, the air pollutants dispersion from the lower ground cannot be taken lightly. The strongest pollutants found in the air are particulate matter (PM), ozone ($O_3$), a combination of nitric oxide (NO) and nitrogen dioxide ($NO_2$) leading to $NO_x$, and sulfur dioxide ($SO_2$), to name a few [9–15]. On the other hand, the diameters of PM smaller than 2.5 μm ($PM_{2.5}$) and 10 μm ($PM_{10}$) have been identified to cause serious health issues [5]. The traffic emission is alone responsible for 39%, 15.8%, and 14.4%, respectively, for $NO_x$, $PM_{2.5}$, and $PM_{10}$ [16]. As a result, air pollutant dispersion modeling focusing on the traffic emission and correlations with the high-rise buildings in a narrow street requires meaningful research [17–20]. A set of four Euro 5-rated diesel vehicles traveling at a constant speed may generated momentary roadside concentrations of $NO_x$ as high as 1.25 mg/m$^3$, with a 25% expected increase for doubling the numbers of vehicles and approximately 50% reduction when considering Euro6-rated vehicles [21]. A real-time investigation has been performed in order to measure formaldehyde (HCHO) emission from one gasoline passenger car, one diesel commercial vehicle, and one diesel heavy-duty vehicle in Europe [22]. The physico-chemical properties of the traffic emission and the density of the air pollutants vary based on the geographic location, time, and space [23,24]. The chemical composition is a variable to define the level to toxicity. There are several possible combinations of reactions such as directly chemical, coagulation as well as condensation of particles and aerosols [24]. The transformation of the pollutants will depend on the local condition such as wind, sunlight, temperature, and humidity, to name a few [24]. Therefore, it would be ideal to understand the air pollutants dispersion on a regional level and further narrow down to the suburb scale with a view to scrutinizing meticulously.

Bangladesh is a country from South Asia with approximately 164.7 million inhabitants [25,26]. Bangladesh loses approximately 80 to 260 million USD annually due to air pollution, which is equivalent to 1.7–7.5% of the city Dhaka's gross product [27,28]. Dhaka is the capital of Bangladesh and the largest city. Dhaka has been subject to rapid urbanization since the late 1990s. However, the city dwellers have been facing severe health issues with below par living standard due to the unplanned urbanization. Due

to the sharp increase in industrial buildings and vehicles near the residential suburbs such as Tejgaon, the air pollution has become one of the challenging tasks to handle. In fact, like other countries, the primary source of air pollution in Dhaka is also due to the combustion of leaded gasoline from the vehicles [28–30]. It has been reported that the professionals working near Tejgaon are exposed to lead (Pb) contamination leading to the Pb concentration in blood between 13 and 132 µg/dL [31], whereas 10 µg/dL is considered to the maximum acceptable concentration described by the WHO [32]. Therefore, the investigations regarding the pollutant dispersion, particularly due to the traffic emission, needs to be further narrowed down to a suburb scale, rather than considering the whole Dhaka city's footprint.

Many developed countries are putting forward big budgets for a sustainable environment and have specific air watch towers to monitor air quality [33–37]. In the atmosphere, the $NO_2$ concentrations reach maximum values in the city center and decrease toward its periphery. Among the inhabitants of Meknes city in Morocco, the respiratory problem reached more acute levels in men than in women, and asthma attacks were noted mainly among women than men in the study period (2010–2014) [38]. Most of the tools they use are based on numerical models, and there are numerous ways of developing such tools. All the methods available have their own advantages and disadvantages in terms of understanding the underlining reasons for mitigating air pollution. Computational fluid dynamics (CFD) has been used to model indoor air quality as well as pollutant dispersion models for decades [39–41]. One major advantage of using a CFD-based dispersion model could be attributed to the sensitivity analyses by varying multifarious geometric parameters within precise resolutions along with the density of the pollutant dispersion to study threshold levels. The majority of the CFD research on pollutant dispersion adopted Reynolds-averaged Navier–Stokes (RANS) [42–45] and large-eddy simulation (LES) [19,39,46,47]. While LES has been attributed to be more efficient in terms of unsteady flows, it has been found to be more reliable on the computational environment and user-defined input parameters [48]. The grid size and characteristics as well as boundary conditions have been found to be more complicated in terms of complex geometries. On the other hand, RANS has been found to be ahead of LES due to having a best practice benchmark and well-defined grid topology [18]. The wind tunnel model presented by Gromke et al. [49] and the Concentration Data of Street Canyons (CODASC) database (https://www.umweltaerodynamik.de/bilder-originale/CODA/CODASC.html, accessed on 18 November 2022) have been widely considered in air pollutant dispersion studies. Furthermore, the atmospheric boundary layer (ABL) can be integrated by a developing boundary layer as a representative. The process can be implemented by ANSYS Fluent$^{TM}$ solver using a user-defined function (UDF) for the calculation of the concentration of air pollutants.

A recent literature survey has suggested that a significant amount of studies focusing on atmospheric phenomena exist. However, most of the studies focus on different geographic locations mostly in well-developed countries where the urbanization progress has been comparatively well-planned. Lauriks et al. [1] investigated the application of RANS to reduce traffic-based air pollution in major roads in Antwerp, Belgium. The study suggests that it is possible to distinguish a level of heterogeneity of the pollutant dispersion based on area of interests. Yuan et al. [50] presented a semi-empirical multilayer urban canopy model to study the traffic emission perpendicularly in a high-density area of Hong Kong and Singapore. Their findings have showcased the benefits of multilayer models for two different geographic locations. The works of Zheng and Yang [3] have provided comprehensive understanding on the advantages and disadvantages of LES and RANS. Recently, Hassan et al. [19] have investigated the air pollutant dispersion in a model street canyon intersection of Dhaka, Bangladesh. The consideration of LES through the finite volume method (FVM) provided both qualitative and quantitative analyses but the study was not conducted on a suburb scale. Therefore, it is still a requirement to understand

air pollutant dispersion based on certain localities, as they vary based on the density of buildings, inhabitants, existence of industrial set-ups, temperature, and wind direction.

The present study aims to investigate air pollutant dispersion in two of the densely populated areas of Dhaka city: namely, Tejgaon and Gazipur. Due to the unplanned urbanization and unpleasant mix up of residential and industrial set-ups, the living standard has been below the standard for many years. The present pollutant dispersion model considers a RANS-based $k - \epsilon$ turbulence approach by the FVM technique with the aid of ANSYS Fluent$^{TM}$. Three distinctive case studies by altering street dimension and traffic emission, based on the actual geographic location (Tejgaon and Gazipur), have been presented. The case studies include: (I) a 2D idealized city model with a straight single street, (II) the effect of a 2D double uniform and non-uniform street, and finally, (III) an irregular multi-street layout based on a real street model. A species transport equation has been solved for carbon monoxide (CO) gas dispersion simulation. The fundamental objective is to gain a deep insight into how precisely the CFD-based FVM-RANS approach can be applied to evaluate the air pollutant from traffic emission in two different regions in a real geographic suburban setup. The key findings of this study will provide more guidelines on possible tool development for a real-time air pollutant monitoring in a suburb-scale based on the type of street and density of vehicles on the streets.

## 2. Mathematical Formulation

The flow properties in the city areas are described mathematically by the mass conservation, momentum, and energy equations. To quantify the turbulent airflow and the pollutants dispersion, the scalar transport equations have been considered.

### 2.1. Continuity, Momentum, and Pollutants Transport Equations

The continuity equation can be written as the following [51]:

$$\frac{\partial \rho}{\partial t} + \frac{\partial u_i}{\partial x_i} = 0 \tag{1}$$

Meanwhile, the momentum equation (Navier–Stokes equation) is the following [52]:

$$\rho \frac{\partial u_i}{\partial t} + \rho \frac{\partial}{\partial x_i} \left( u_i u_j + \overline{u_i u_j} \right) = -\frac{\partial P}{\partial x_i} + \tau_{ij} \tag{2}$$

Finally, the pollutants transport equation can be written as below [53]:

$$\frac{\partial C^\alpha}{\partial t} + u_j \frac{\partial C^\alpha}{\partial x_i} = \frac{\partial}{\partial x_i} \left( (D^\alpha + \frac{\nu_t}{S_{ct}}) \frac{\partial C^\alpha}{\partial x_i} \right) \tag{3}$$

where $u_i$ and $P$ are the Reynolds average velocity and pressure, respectively, $u_i^{'}$ is the fluctuating velocity, $\rho$ is the fluid density, $\nu$ is the kinematic viscosity, $C^\alpha$ is the concentration of pollutant species $\alpha$, $D^\alpha$ is the diffusivity, $\nu_t$ is the turbulence eddy viscosity, $\tau_{ij}$ is the share stress and $S_{ct}$ is the turbulence Schmidt number. The term $\overline{\rho u_i u_j}$ is the time-averaged rate of momentum transfer due to turbulence.

### 2.2. $k - \epsilon$ Turbulence RANS Model

The $k - \epsilon$ RANS-based dispersion model is used to determine the turbulent velocity where the pollutants dispersion equation is coupled with the $k - \epsilon$ model. The transport equations for the turbulent kinetic energy (TKE) and dissipation rate $\epsilon$ are presented next.

The k-equation is expressed as [54]:

$$\frac{\partial (\rho k)}{\partial t} + \frac{\partial (\rho k u_i)}{\partial x_j} = \tau_{ij} \frac{\partial \overline{u}_i}{\partial x_j} - \rho \epsilon + \frac{\partial}{\partial x_j} \left( (\mu + \frac{\mu_t}{\sigma_k}) \frac{\partial k}{\partial x_j} \right) \tag{4}$$

Meanwhile, the $\epsilon$-equation is written as the following [54]:

$$\frac{\partial(\rho\epsilon)}{\partial t} + \frac{\partial(\rho\epsilon u_i)}{\partial x_i} = C_{\epsilon_1}\frac{\epsilon}{k}\tau_{ij}\frac{\partial\overline{u}_i}{\partial x_j} - C_{\epsilon_2}\frac{\rho\epsilon^2}{k} + \frac{\partial}{\partial x_j}((\mu + \frac{\mu_t}{\sigma_\epsilon})\frac{\partial\epsilon}{\partial x_j}) \tag{5}$$

where $\nu_t = \frac{C_\mu \rho k^2}{\epsilon}$, $C_\mu = 0.09$, and $\sigma_k = 1$ $\sigma_\epsilon = 1.3$, $C_{\epsilon_1} = 1.44$ and $C_{\epsilon_2} = 1.92$.

### 2.3. Boundary Condition

The power-law formula as an inlet velocity profile has been considered to model the wind velocity profile in the ABL, which can be written as the following [55]:

$$u(y) = 4.7(\frac{y}{0.12})^{0.3} \tag{6}$$

Meanwhile, the turbulent kinetic energy and dissipation rate profiles are specified as following [56]:

$$k = \frac{u^{*2}}{\sqrt{C_\mu}}(1 - \frac{y}{\delta}) \tag{7}$$

$$\epsilon = \frac{u^{*3}}{ky}(1 - \frac{y}{\delta}) \tag{8}$$

where $u$, $y$, $\delta$, $u^*$, and $k$ are the vertical velocity profile, vertical distance, boundary layer depth, frictional velocity, and Von Karman constant, respectively, with $C_\mu$ being 0.09.

No slip boundary conditions have been used on the boundary walls. The upper boundary has considered a symmetry boundary. The dispersion's source is mainly related to the traffic volume in Bangladesh. Initially, we have considered the emission rate of the CO to be about 10 g/s, which is equal to that in the reference [17]. The source of the CO has been considered at the ground level, as shown in Figure 1.

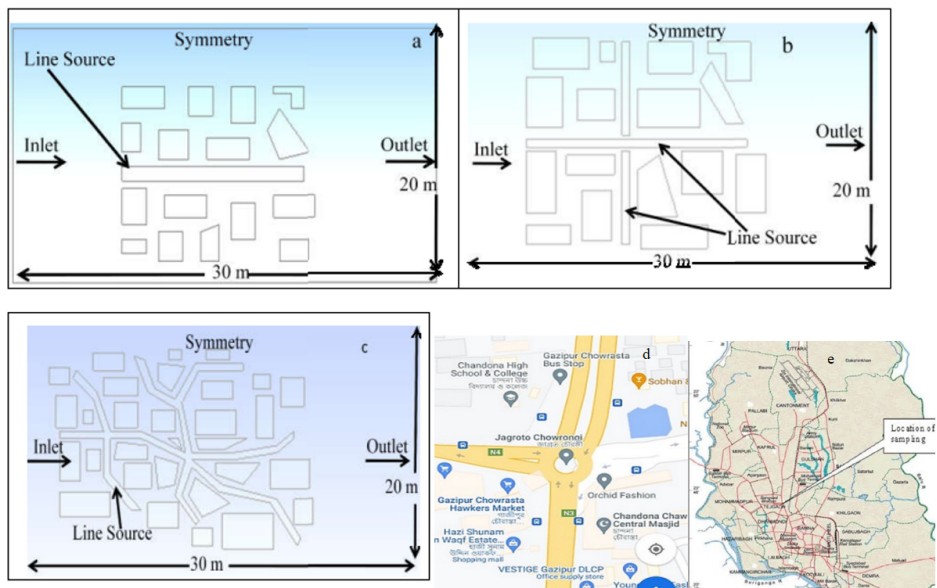

**Figure 1.** Geometric configuration and boundary conditions for two intersections of Gazipur chowrasta in (**a**–**c**) the city model for the irregular street configuration of Tejgoan area in Dhaka city, (**d**,**e**) represent Google Map snippets obtained on 10 October 2021 of two roads intersection of Gazipur chowrasta interections and Tejgoan industrial area, respectively.

*2.4. Turbulent Kinetic Energy*

Among all parameters, turbulent kinetic energy (TKE) is also a major variable for the dispersion of the pollutants. Physical TKE is obtained by components of velocity fluctuation. Typically, it can be measured by the averaged of turbulence normal stress [57,58]

$$TKE = \frac{1}{2}(\overline{u}^2 + \overline{v}^2 + \overline{w}^2). \tag{9}$$

## 3. Methodology

### 3.1. Measurement Site, Wind Direction, and Seasonal Impact

The area of interest has a length of 200 m and width of 150 m with irregularly structured buildings on both sides of the street canyon. The length-to-width ratio of the street is 1.33 m. Different types of fuel-driven vehicles such as buses, trucks, and cars are on the streets on a regular basis. An average of thirty thousand vehicles/week move in both the Gazipur chowrasta (Chowrasta is a Bengali word, meaning the point where four streets meet) intersection and the Tejgoan industrial area. The average vehicle speed at chowrasta is about 4.5 km/h, which is less than the speed of a typical healthy person's walking speed. The slow mobility could be attributed to the high-density of traffic on the narrow streets. On the other hand, the vehicle's speed at Tejgoan is about 6.4 km/h [57]. Typically, the wind blows in Dhaka city with its vicinity in the northwest direction in winter season, and during in the summer or monsoon season, it blows in the south or southeast direction. The wind's speed exhibits a strong seasonal cycle. For instance, in the March–August period, it has higher wind speed than that in the September–February period. The wind speed also exhibits a diurnal cycle, and in the afternoon, the speed becomes stronger compared to that at night. The normal frequency of the wind velocity is about 2/s, and it increases approximately 42% higher in the monsoon season. Meanwhile, the frequency becomes around 9% less than its normal value during the winter season.

### 3.2. Computational Domain Discretization

The methodology consists of solving the transient, Reynolds-averaged, mass, and momentum conservation equations for the mean flow fluid. The mass fraction conservation equation for the dispersion of pollutants has been used to reach the steady-state condition. For the boundary conditions, an inlet velocity profile for the inflow boundaries, and zero gradients of pressure outlet for the outflow boundaries are considered. The computational domain represents the model with 30 m × 20 m (length × width) as presented in Figure 1. Figure 1a,b depict the regular street configuration of Gazipur chowrasta, and the irregular street configuration of Tejgaon industrial area has been presented in Figure 1c with the inlet and outlet placements for a better understanding. The images from Google Map (obtained on 10 October 2021 have been placed in two different frames to show the street configurations of Gazipur chowrasta (Figure 1d) and the Tejgaon area (Figure 1e), respectively.

The discretization of computational domain is a significant step in the pollutant dispersion modeling through CFD as it ensures both the numerical stability and validation of the data. The domain is subject to discretization horizontally, as shown in Figure 2. The grid has a higher spatial resolution, and the spatial concentration distributions near the measurement point to analyze wind flow properly. Figure 2a–c demonstrate the mesh generation of the considered street as described in Figure 1a–c. The velocity distribution is at the center of the model, based on Figure 1a. Three different face sizes have been applied to implement ANSYS meshing. Face sizing is an intuitive method applied on any specific face or surface to control the mesh size including mesh transition on a specified growth rate. The velocity distribution by altering face size has been shown in Figure 2d as a sample based on Figure 1a.

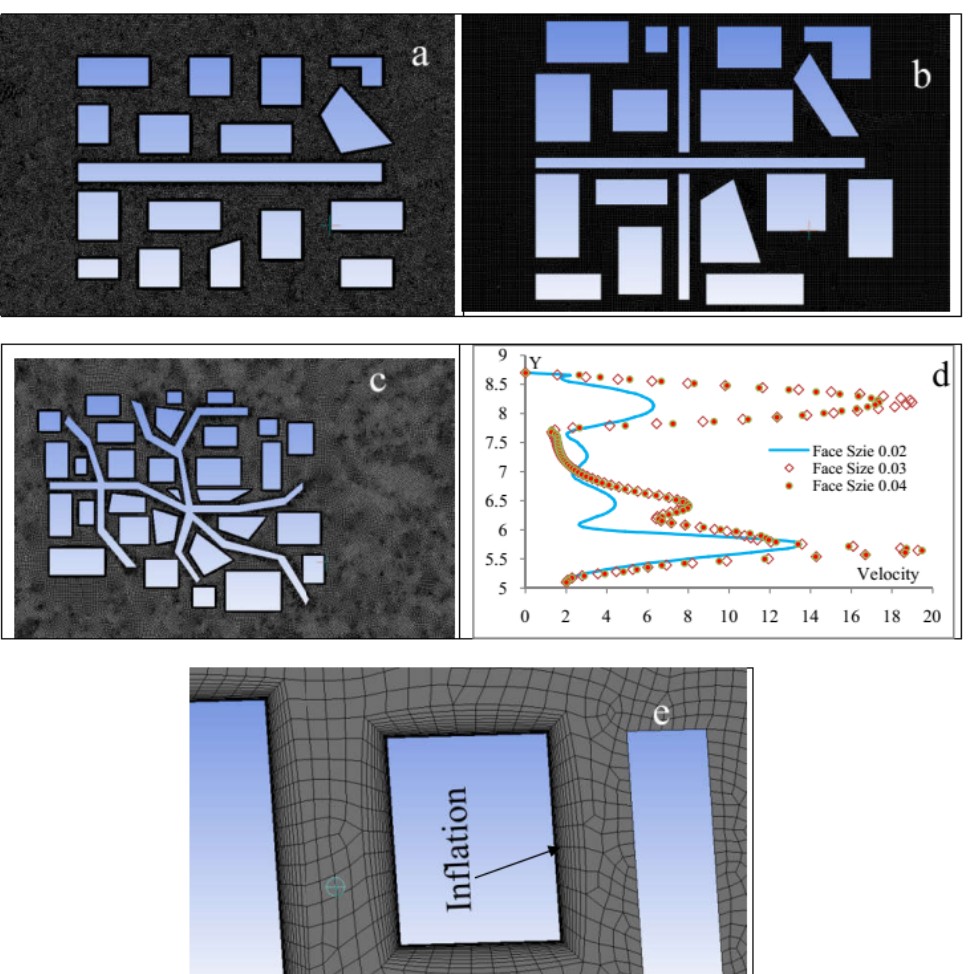

**Figure 2.** Mesh generation of the city model for the regular street configuration in Gazipur chowrasta intersections: (**a–c**) the mesh of the city model for the irregular street configuration of Tejgoan area in Dhaka city; (**d**) velocity distribution at the center of the model; (**e**) closer view of the grid of the building based on the model in Figure 1a.

### 3.3. Model Development and Variables

The moving vehicles in urban areas used fossil fuels that act as the source of air pollutants (CO). The exhausted pollutants that pass through the street canyon are oxidized products such as CO, carbon dioxide ($CO_2$), nitrogen oxide(s) ($NO_x$), unburden hydrocarbons, the volatile organic compound (VOCs), suspended particulate matters (SPM), and aerosols. $CO_2$, water vapor, and SPM are the main products of the complete combustion of the fuel. The CO is produced from incomplete combustion of the fuel due to higher temperature generation inside the combustion chamber of engines. People who live near the main road suffer from cerebral shrinkage due to the negative impact of CO. CO gas has been considered as a source of air pollutant gas in numerical model simulation. As mentioned earlier, the well-accepted $k - \epsilon$ turbulence model in the RANS (Reynolds-averaged Navier–Stokes) approach has been considered. This allowed us to investigate the inner size of the canyon near the buildings of the wind flow and the turbulent pollutants dispersion. The source of pollutants is considered normal to the street that is linear to the flow of wind. The Atmospheric Boundary Layer (ABL) has been implemented in the ANSYS Fluent$^{TM}$ solver using a user-defined function (UDF) to calculate the concentration of the pollutant.

### 3.4. Model Validation and Grid Independence

Model validation is performed prior to case studies. The current model is validated against the wind tunnel experiment's data from the study of Gromke and Ruck [59,60]. The

validation plot against the CODASC experimental results has been presented in Figure 3, showing the agreements in terms of inlet velocity profile, as shown in Figure 3a,b inlet turbulent kinetic energy (TKE). In addition, the normalized pollutant concentrating contours inside the canyon on both sides at leeward (wall A) and windward (wall B) in Figure 4a–c respectively. This model also has been validated against CODASC experimental results. It could be seen that the normalized concentration in leeward (wall A in Figure 4a) is approximately 75% higher than that of windward (wall B in Figure 4a). A similar visualization is obtained in terms of the RANS model as well as shown in Figure 4b,c. Meanwhile, Figures 5 and 6 depict the comparison between the present RANS experimental and the wind tunnel experimental results. A quantitative presentation of the pollutant concentration has also been executed at five various span-wise positions on both the leeward (wall A) and windward (wall B) as Z/H = 0; Z/H = 1.66; Z/H = 3.334; Z/H = −1.66; and Z/H = −3.334. Based on the demonstrations from Figures 5 and 6, the current numerical simulation results have good agreement with the CODASC experimental results.

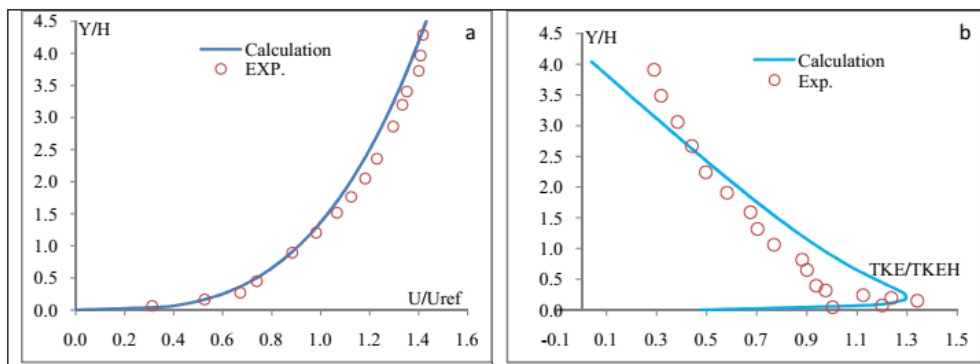

**Figure 3.** (**a**) Comparison of the inlet velocity profile with the CODASC experiment and (**b**) Turbulent Kinetic Energy (TKE) profile with the CODASC experiment.

The grid independence test (GIT) is an important factor to establish numerical simulations. For the validation of the present study, GITs have been carried out using several grid sizes. Some of the relevant configurations include: (I) maximum face size for regular single street = 0.03, nodes = 218,020, and elements = 215,526; (II) maximum face size for irregular single street = 0.03, nodes = 40,186, and elements = 62,647); (III) maximum face size for regular double street = 0.03, nodes = 59,051, and elements = 57,544; (IV) maximum face size for irregular double street = 0.03, nodes = 56,305, and elements = 88,120); and (V) maximum face size for irregular multi-street = 0.03, nodes = 215,909, and elements = 211,759. A finer grid has been generated near the building compared to the far-field that will work to capture the boundary layer phenomenon. For all grid sizes, the velocity distribution remains the same and matches the experiment. A similar velocity distribution was observed at the center of the street canyon. This simulation has been executed considering grid 0.03.

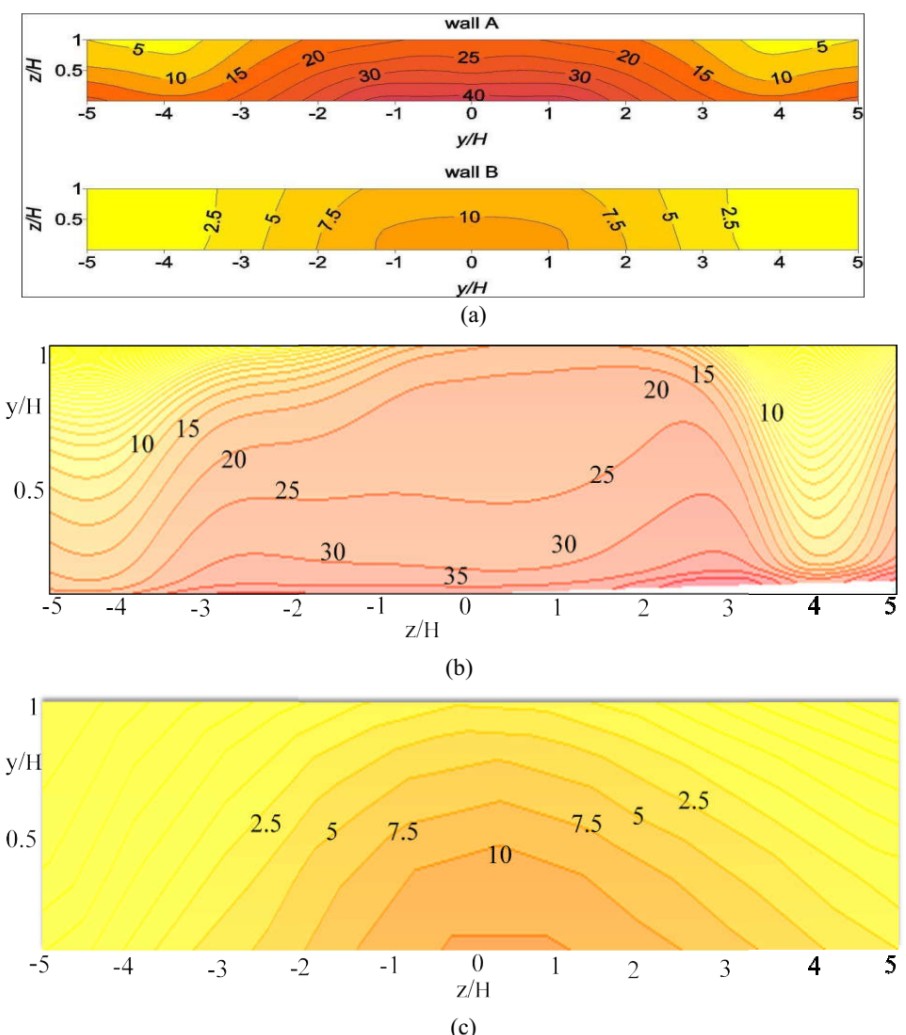

**Figure 4.** The wind tunnel experimental time-averaged normalized pollutants concentration presenting in (**a**) on both wall A (Leeward) and wall B (Windward); (**b**) Leeward (wall A) for presenting RANS; and (**c**) wall B (Windward) for presenting RANS.

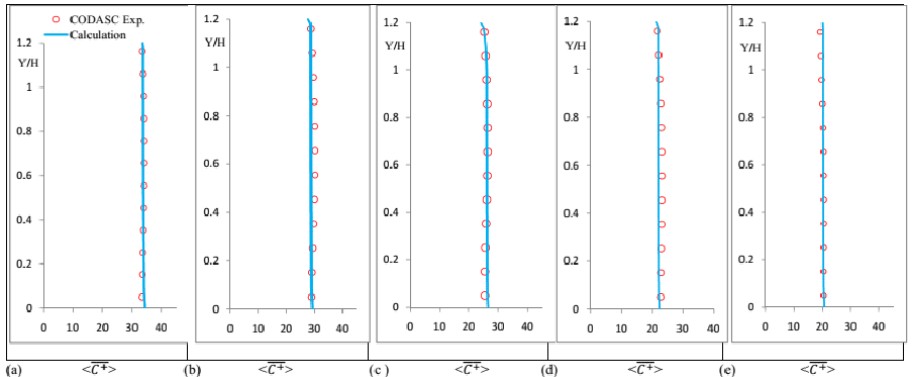

**Figure 5.** Time-averaged normalized pollutant concentration profiles at five separates spanwise positions on Leeward (wall A) (**a**) Z/H = 0; (**b**) Z/H = 1.66; (**c**) Z/H = 3.334; (**d**) Z/H = −1.66; and (**e**) Z/H = −3.334 to compare current RANS vs. WT Experiment.

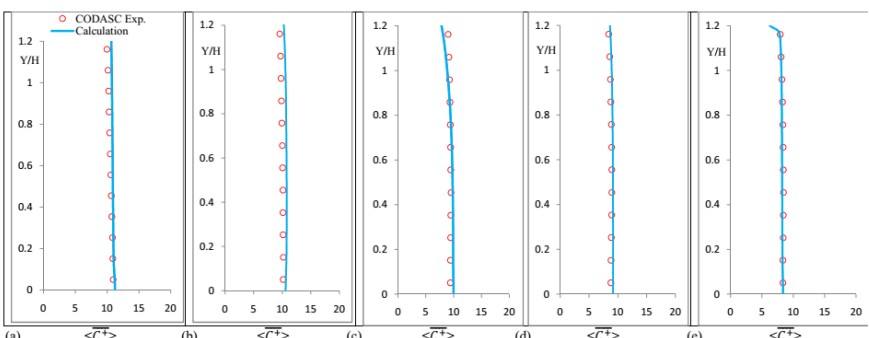

**Figure 6.** Time-averaged normalized pollutant concentration profiles at five separates spanwise positions on Windward (wall A) (**a**) Z/H= 0; (**b**) Z/H = 1.66; (**c**) Z/H = 3.334; (**d**) Z/H = −1.66; and (**e**) Z/H = −3.334 to compare current RANS vs. WT Experiment.

## 4. Simulated Results

The obtained results were developed to understand the impact of type and dimension of the streets, traffic emissions, and wind attributes. The empirical demonstrations have been divided into case studies and separate discussions at the later stage to imply the key findings of this study.

### 4.1. Case Studies on the Street Geometries

Three different case studies have been presented to understand the influence of the types of street based on the real street design of Dhaka city. Case studies I, II, and III represent the impact of single streets, double-crossed streets, and multiple irregular streets, respectively. The purpose of case studies is to identify the air pollutants dispersion in different geometries of the streets, which were inspired by the real street layouts of Tejgaon and Gazipur.

#### 4.1.1. Case Study I: Influence of Single Street

In this part of the case study, simulation results for aspect ratio (L/W) with fixed wind directions with various lengths and shapes of the streets have been presented. There are various kinds of streets as lengths L = 10 m, L = 15 m, and L = 20 m have been considered for both regular and irregular streets. The concentration contours are depicted in terms of the volume fraction of carbon monoxide (CO) in the air. The velocity contour and the CO mass fraction concentration are shown in Figure 7.

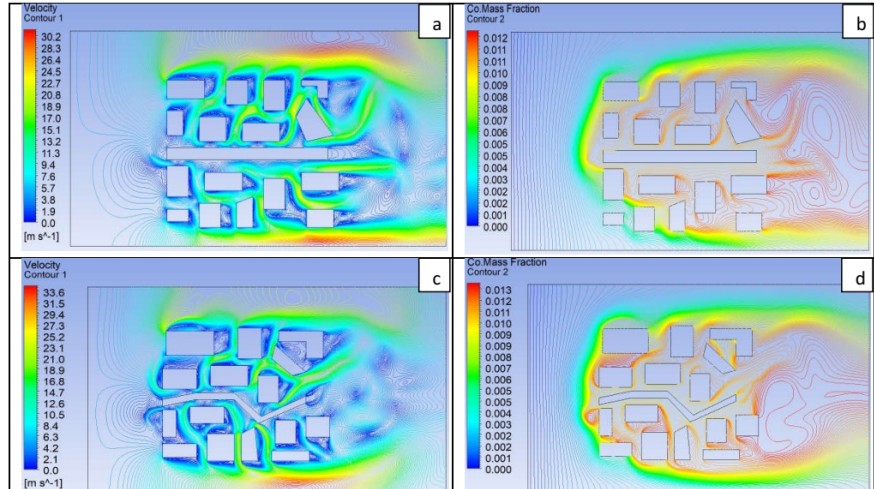

**Figure 7.** (**a**) Velocity contours for a single regular street as well as a single irregular street in (**c**), and the carbon monoxide (CO) mass fraction for a single regular street (**b**) and (**d**) single irregular street.

The velocity contours for a single regular street and a single irregular street are shown in Figure 7a and Figure 7c, respectively. It could be observed that the highest velocities have been obtained to be 30.2 m/s and 33.6 m/s, respectively, for a single regular street and irregular street. However, the surroundings of the single regular street are found to have experienced more wind velocity due to the more degree of freedom. In addition, the irregular single street created hindrance to a greater extent, and hence, the velocity contour exhibited a less dense profile. Inlet velocity profiles are kept consistent, and the velocity profiles near the outlet have been observed and the influence of the type of single streets could be understood. To summarize, it has been found that the transformation from a single regular street to a single irregular street could increase the peak value of the velocity by 11.26%.

In another part of this case study, the mass fractions of CO have been shown in the corresponding street types. It has been observed that the peak mass fraction of CO is 0.012 in terms of a single regular street (Figure 7b) and 0.013 in terms of a single irregular street (Figure 7d). The mere 8.33% increase in the maximum CO mass fraction could be described in the light of the geometry of the street. The regular single street created more free space for the airflow, whereas the irregular street contained more obstacles in the mid-section, leading to a slight increment in the pollutant concentration overall. However, the mass fraction of CO has been found to be higher for a single regular street near the outlet, which was between 0.010 and 0.012. This observation could be explained in terms of the presence of the traffic and the traffic emission.The traffic in Dhaka city near Tejgaon or Gazipur moves comparatively slower due to the high density of the vehicles in a narrow street. Therefore, a longer street will accommodate more vehicles, and consequently, the pollutant concentration near the outlet will increase and continue to spread due to the continuous exhaustion of traffic.

### 4.1.2. Case Study II: Impact of Double-Crossed Streets

Case study II aims to investigate the impact of double-crossed streets in both regular and irregular dimensions, as defined before. The simulated results have been presented in Figure 8a,c to demonstrate the velocity contours, and Figure 8b,d depicts the distribution of CO mass fraction across the geometries for regular and irregular streets, respectively.

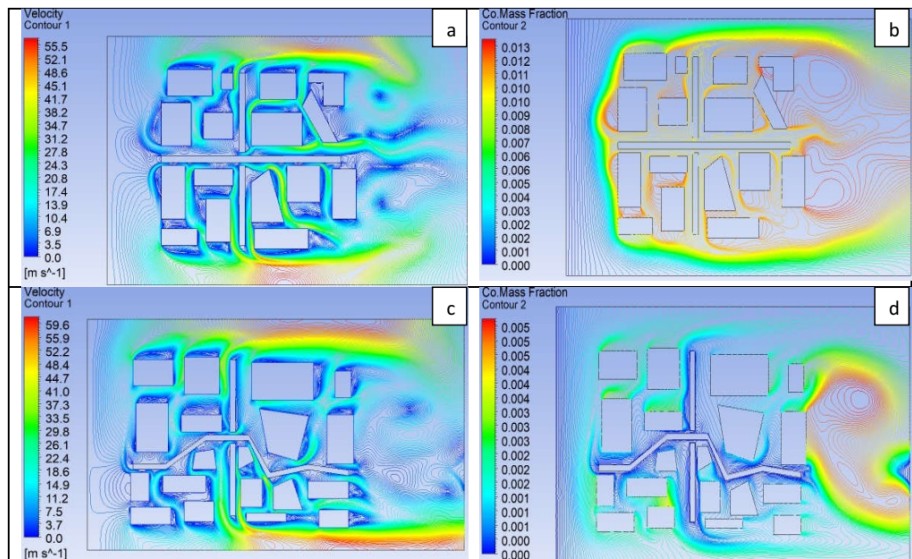

**Figure 8.** (**a**) Velocity contours for double-crossed regular streets and double-crossed irregular streets as shown in (**c**). The carbon monoxide (CO) mass fraction for double-crossed regular streets and irregular streets are shown in (**b**,**d**), respectively.

As per Figure 8a,c, the peak velocity values increased significantly for both regular and irregular streets than those of single streets, as described in Figure 7a,c. The peak

velocity values have been recorded to be 55.5 m/s and 59.6 m/s for double regular and irregular streets, respectively. It should be highlighted here that the maximum wind velocities have been found to be 30.2 m/s and 33.6 m/s for single regular and irregular streets, respectively. Therefore, it could be observed here that by increasing the dimension, particularly the length and altering the design of the streets as crossroads instead of a long single road, the wind velocity will exhibit an increased peak value. For double regular and irregular streets, the peak values of the wind velocity increased approximately 83.77% and 77.38%, respectively. The existence of the crossroads in both regular and irregular streets created alternative pathways for the airflow, and due to increased wind velocity, the outlet experienced re-circulations, which was qualitatively higher at the end of irregular street. Furthermore, the wind velocity contours also suggested that the irregular street (Figure 8c) experienced greater velocity in both windward and leeward sides (between 44.7 and 59.6 m/s).

On the other hand, double regular crossroads had an insignificant impact on the CO mass fraction, as shown in Figure 8b. There is a slight increase in the maximum CO mass fraction value compared with that of a single regular street. While the peak CO mass fraction has been recorded to be 0.012 for a single regular street, the highest CO mass fraction increased approximately 8.33% to 0.013. Therefore, the consideration of up-sizing the length of the single road and altering to a double crossroad lowered the air quality. This behavior could be explained in terms of two circumstances. First off, the increased dimension allowed more traffic to ply on the road, leading to a greater release of pollutants. Secondly, the role of the crossroad did not create much hindrance comparatively due to the improved peak wind velocity. The 83.77% increased peak velocity overpowered the idea of considering a regular double crossroad. Meanwhile, there is a significant decline in the maximum CO mass fraction in terms of irregular double streets (0.005), which is 0.013 in terms of single irregular street. The 61.54% decline in the CO mass fraction could be attributed to the geometry of the double crossroad, as shown in Figure 8d. Due to the increasing obstacles, the pollutants from the traffic do not have an increased degree of freedom to disperse along the wind velocity. However, once the pollutants reached the close proximity of the outlet, the density of the pollutants increased significantly due to surpassing all the hindrance from the crossroads and buildings. As a consequence, a high concentration of CO mass fraction as well as dense vorticities have been observed near the outlet.

### 4.1.3. Case Study III: Effect of Multiple Irregular Streets

The current study focuses on Tejgaon and Gazipur, which are two of the busiest places in Dhaka. While case studies I–II demonstrated the impact of single and double regular and irregular streets on the pollutants dispersion as well as wind velocity, they do not represent the actual geometries of the streets of Tejgaon and Gazipur. The majority of the streets in those two locations consists of unplanned buildings and multiple narrow streets. Therefore, case study III has focused on multiple irregular streets as part of the final case study of this scheme.

The transformation from single irregular streets to double irregular streets displayed a significant rise in the peak velocity (83.77%). Therefore, it is anticipated that further up-sizing the double irregular streets to multiple irregular streets would augment the peak velocity rapidly. Figure 9a confirmed this hypothesis, where the peak velocity was obtained to be 95 m/s. This led to a sharp rise of 59.4% and 182.74% in peak velocity from that of double and single irregular streets, respectively. This outcome could be described in the light of geometrical changes in the shape of street, which added more obstacles to the wind velocity in between. Therefore, once the air approached near the outlet, the velocity increased rapidly and created multifarious batches of vortices. The vortices are more dense near the smaller free space of the obstacles due to the turbulence and later spread throughout the outlet zone.

Due to the increased obstacles, the pollutants dispersion will be comparatively lower. Figure 9b suggests that the peak value of CO mass fraction is 0.013, which is an 160% increase. However, comparing to the outcomes from the single irregular street as reported in Figure 7d, this increasing pollutants is an improvement. As per Figure 7d, the maximum CO mass fraction is 0.013, but the peak wind velocity has been recorded to be 33.6 m/s (Figure 8a). For the multiple irregular streets, the CO mass fraction is still 0.013 but with an improved peak wind velocity of 95 m/s. Therefore, the wind will be able to divert the pollutants far away from the localities to the outlet zone efficiently. Since the wind velocity varies frequently inside the city area, many vortices have been formed at the end of the street canyon. At the entrance of the street, the concentration of the pollutants is higher on the leeward side than those at the concentration on the windward side of the buildings. In the middle of the street canyon, low wind velocity has been observed. This is why the multi-irregular streets work as barriers to the dispersion of the pollutants that are exhausted from the sources, particularly traffic.

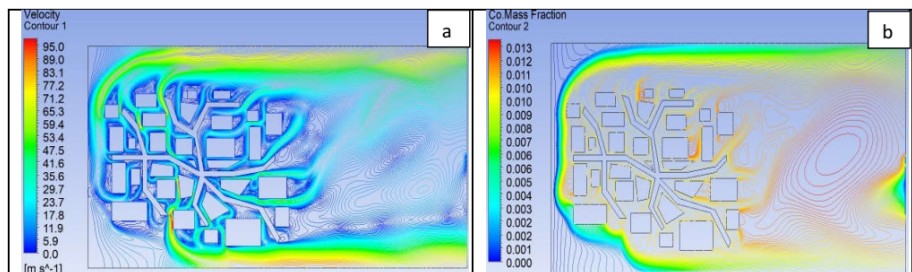

**Figure 9.** (**a**) Velocity contour, and (**b**) carbon monoxide (CO) mass fraction for multiple irregular streets.

### 4.2. Impact of Wind Velocity and Direction

Wind plays a significant role in spreading the pollutants. The impact of street cannot alone influence the type of air quality of an area. Dhaka is a densely populated country, and the streets are filled with different types of vehicles. Therefore, the diversion of the traffic emission will depend significantly on the wind velocity and direction. The present study considers single regular and irregular streets, double regular and irregular streets, and multilayer streets. Therefore, the impact of wind velocity has been studied in all possible combinations of streets considered in this research. The following sections aim to investigate the impact of wind.

The effect of wind velocity under different layouts of the street has been presented in Figure 10. The purpose of examining the influence of wind velocity could be attributed to the formation of vorticities. In general, wind should form a greater amount of vortices after overcoming the outlet zone. However, the type of circulation and shape would differ based on the type of obstacles wind faces between the inlet and outlet.

Figure 10a,b depict the effect of wind velocity in the presence of a single regular and irregular street, respectively. The wind faced fewer obstacles across the path between the inlet and outlet in terms of a single regular street. Therefore, the sharp increase in peak wind velocity was not noticed overall. Wind crossing across the outlet in a single regular street demonstrated a peak velocity of 31.1 m/s, which increased 84.24% to 57.3 m/s when a single irregular street is present. This rapid augmentation in the peak velocity could be attributed to the geometry of the street where the the wind faced hindrance over the single irregular street and therefore rapidly formed batches of vortices near the outlet. Therefore, it is expected that the dispersion of pollutants will have comparatively lower mobility in terms of a single irregular street.

Meanwhile, Figure 10c,d demonstrate the impact of velocity of the wind in terms of double regular and irregular streets concurrently. The highest velocity has been observed to be around 34.7 m/s for a double regular street, whereas it was 61.5 m/s for a double irregular street, which increased around 77.23%. However, larger shapes of turbulence

could be observed in both Figure 10c,d, where Figure 10d exhibited a greater shape. This could be explained in the similar fashion as Figure 10a,b, as the wind faced further obstacles between the inlet and outlet. Therefore, as the wind surpassed all the structural buildings, the velocity experienced more free space, which led to swift augmentation in the velocity including the formation of turbulence.

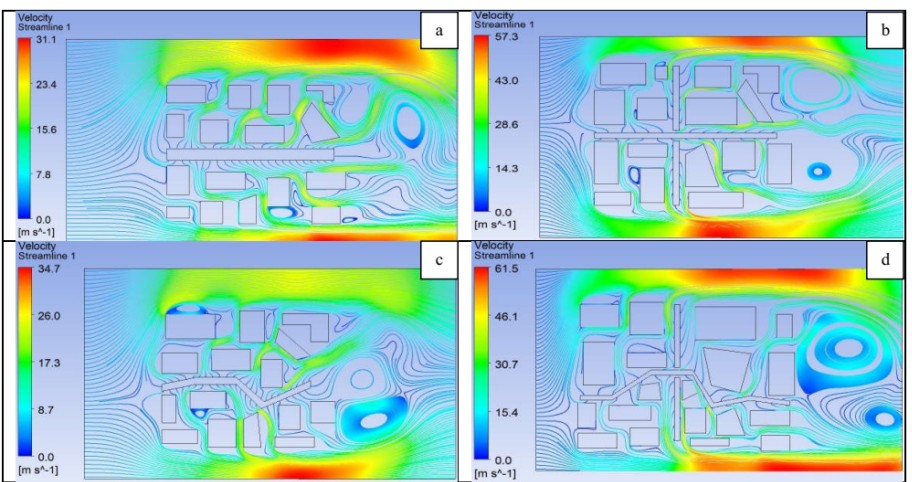

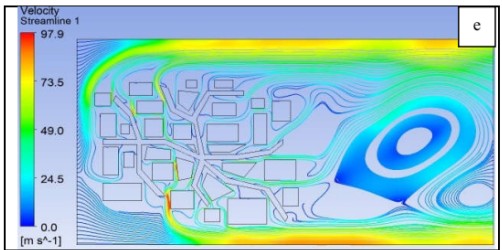

**Figure 10.** Variation of wind velocity in terms of five different streets: namely, (**a**) single regular, (**b**) double regular, (**c**) single irregular, (**d**) double irregular, and (**e**) multiple irregular.

Finally, the influence of wind velocity and associated visualization could be seen in Figure 10e in terms of multiple irregular streets. The outcome is consistent with the hypothesis stated earlier. Due to the existence of more complicated geometry, there are more buildings within narrow streets. Furthermore, the number of traffic will be higher, leading to more exhaustion. The CO mass fraction has been investigated already and has not been rehearsed in this section. The vortices observed in Figure 10e had the largest shape among the types of streets considered in this study. The peak velocity of 97.9 m/s suggests that the wind velocity further increased after crossing the outlet. Therefore, it could be noticed from this part of the analysis that as wind faces more obstacles between the inlet and outlet, the velocity will increase further near the outlet or after over passing all the obstacles. The intensity of the obstacle will directly influence the shape of turbulence caused by the wind velocity.

### 4.2.1. Effect of Wind Direction and Seasonal Cycle

Pollutants dispersion is significantly affected by the variability of wind direction. In this section, the simulated results have been presented in different street geometries in several wind flow directions. In Dhaka city, the wind flow pattern and speed change with the change of the seasons. For example, during the winter season, the wind blows with low velocity from September to February in the northwest direction, and it exhibits a strong seasonal cycle shown as presented in Figure 11. On the other hand, in the summer season, the wind blows with an augmented mobility during March–August in the south–southeast direction, and it showcases a strong seasonal cycle shown in Figure 12. Hypothetically, the

wind speed starts to plummet at night, and the concentration of pollution is mitigated due to the comparatively lower traffic density on the streets.

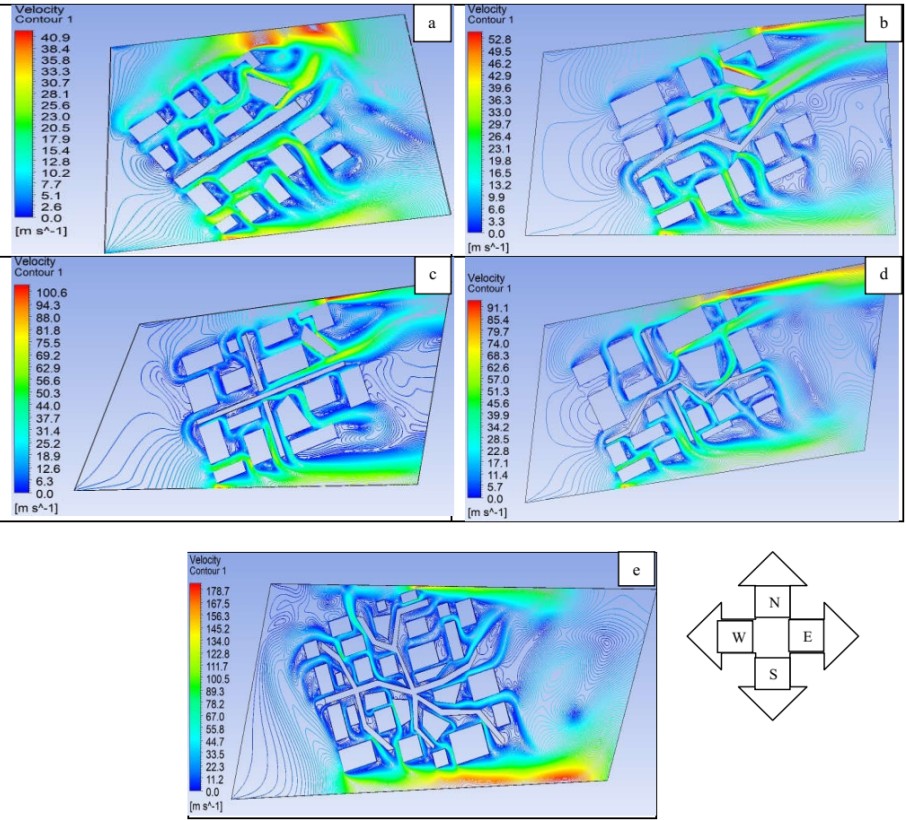

**Figure 11.** Variation of wind velocity from northeast wind direction in terms of five different streets: namely, (**a**) single regular, (**b**) single irregular, (**c**) double regular, (**d**) double irregular, and (**e**) multiple irregular.

Figure 11a,b pinpoint the effect of wind direction in terms of elevating the wind velocity in the northwest direction for single regular and irregular streets, respectively. As mentioned earlier, this part of analysis considers the generic scenario of the winter season in Bangladesh focusing on the type of street in Tejgaon and Gazipur. It could be observed that the peak velocity in a single irregular street was quantitatively 29.09% higher than that of a single regular street. However, the density of augmented velocity has been visible mostly surrounding the northern region for a single regular street (Figure 11a), which is less dense in terms of a single irregular street (Figure 11b. As the wind direction was being considered from the northwest direction, the wind experienced more free space before entering the assigned inlet zone. Therefore, the velocity is found to have increased values near the northern zone. However, single irregular streets have more obstacles for the wind and could occupy more vehicles on the streets. As a result, even the peak velocity has been recorded to be higher for a single irregular street, and the dispersion would be slower regardless of the wind direction. Furthermore, in terms of double regular and irregular streets (Figure 11c,d), the peak velocities are expected to increase, which could be seen. However, the maximum velocity has been found to be 100.6 m/s and 91.1 m/s for double regular and irregular streets, respectively. The outcome suggests that the peak velocity decreased approximately 9.44% for double irregular streets, whereas the peak velocities are always higher for double irregular streets during the overall comparison in Figure 10c,d. This could be explained in terms of wind directional analysis from the northwest, which is closer to the inlet, and double irregular street geometries narrowed down the free space for the wind due to the structural buildings and more traffic occupancy. Therefore, the peak wind velocity as well as the overall distribution of air experienced a reduction in terms of

double irregular streets. In addition, the existence of multi-irregular streets will compress the inlet wind further; however, the wind would be able to find an alternative way toward the southeast direction to bypass the way toward the outlet. As a result, the augmented peak velocity of 178.7 m/s has been recorded for a multi irregular street, as presented in Figure 11e. The 96.16% augmentation in the peak velocity suggests that despite the initial hindrance for the wind from the inlet, the wind is able to overcome the barrier to go toward the outlet without demonstrating higher velocity in the mid-section, which is the area for the inhabitants.

The influence of wind velocity from the south or southeast direction has been presented in Figure 12. During the summer, the average temperature of the weather is elevated, and the wind mostly flows from southwest or south direction. Therefore, it has been anticipated that the peak velocity values will be increased due to the direction of the wind being considered far from the inlet. The wind will be able to find the alternative pathway through the southeast direction due to the free space and reach near the outlet zone through the structural buildings (both residential and industrial).

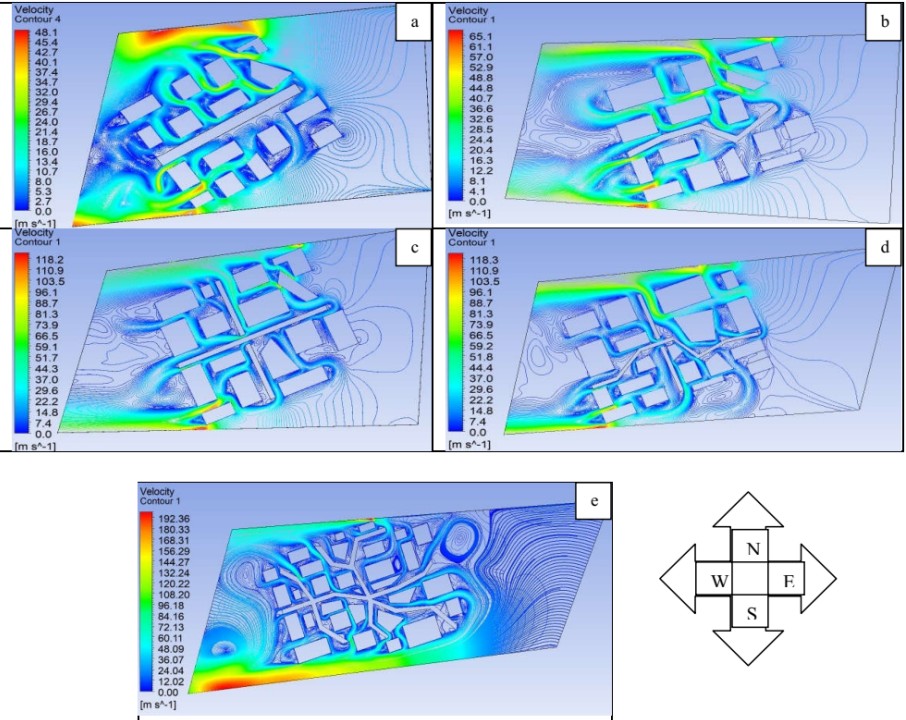

**Figure 12.** Variation of wind velocity in terms from south/southeast wind flow of five different streets: namely, (**a**) single regular, (**b**) single irregular, (**c**) double regular, (**d**) double irregular, and (**e**) multiple irregular.

Figure 12a,b demonstrate the impact of wind velocity from the southeast direction for single regular and irregular streets, sequentially. It could be observed that the density of the wind is comparatively higher near the northern and southern zones due to the proximity of the inlet and wind direction, respectively. However, the peak velocity for both single regular and irregular streets augmented to 48.1 m/s (17.60% higher than maximum velocity of the northeast) and 65.1 m/s (23.30% greater than the peak velocity of the northeast) due to the changes in the wind direction. Moreover, the influence of this underlying wind direction in terms of both double regular and irregular streets had an identical effect with peak velocity recorded between 118.2 and 118.3 m/s. This could be attributed to the fact that despite the type of street (regular or irregular), double crossroads do not create additional free space in between, and due to the changes in the wind direction, wind velocity will face similar types of the obstacles from the inlet passing from the southeast direction. However, as irregular multi streets have been considered, due to the free space near the southeast region

and elevated velocity, the wind will create larger vortices near the obstacle before flowing toward the outlet. Due to the turbulence, the peak velocity will rise quickly, leading to 192.36 m/s, which was a 7.644% increase from Figure 11e.

### 4.2.2. Influence of Wind Direction in CO Mass Fraction Distribution

The observation of CO mass fraction represents mostly the pollutant caused from the traffic emission. The role of wind direction in terms of velocity has been discussed already. This section aims to analyze the effect of wind direction in terms of traffic-generated pollutants dispersion. It has been expected that due to the wind direction from the northeast region in the winter, the maximum CO mass fraction, i.e., the highest amount of pollutants will be almost consistent regardless of the type of street due to the increased obstacle and the wind direction being in proximity of the inlet. However, the distribution of pollutants will vary based on the physical dimensions of the streets.

Figure 13 depicts the overall distribution of CO mass fraction as the wind flew from the northwest direction. It could be observed that the peak CO mass fraction always remained 0.012 regardless of the type of street. There could be a few changes at an increased resolution, but this will not impact the original hypothesis presented earlier. However, the distribution of CO mass fraction has been found to be more dense in terms of a single regular street, as shown in Figure 13a, due to there being fewer obstacles, which were significantly reduced for single crossroads in Figure 13b. Therefore, it could be a potential health hazard for the inhabitants dwelling or traveling near the single regular street near the winter season in Tejgaon and Gazipur areas. Similarly, the density of the pollutants is comparatively higher near the residential and industrial areas for the double regular street (Figure 13c) compared with that of a double irregular street (Figure 13d). Meanwhile, the wind mostly diverted the pollutants through the southeast direction due to increased velocity in terms of the multiple irregular streets as shown in Figure 13e toward the outlet, leaving the inhabitants less exposed to the traffic-generated pollutants.

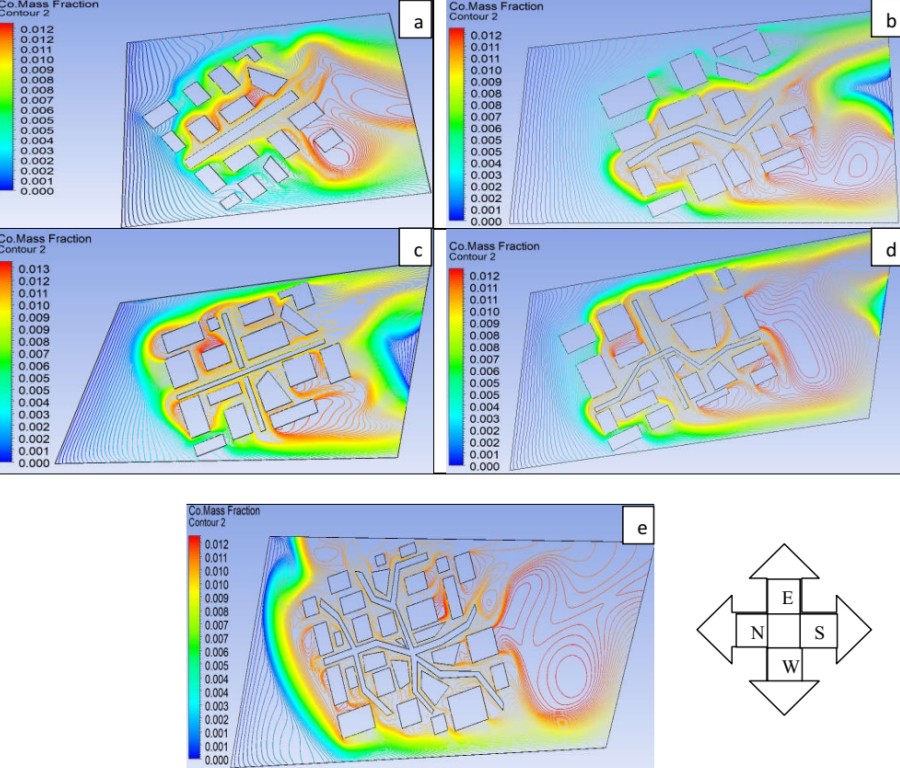

**Figure 13.** Distribution of traffic-generated CO mass fraction in terms of northwest wind flow of five different streets: namely, (**a**) single regular, (**b**) single irregular, (**c**) double regular, (**d**) double irregular, and (**e**) multiple irregular.

Meanwhile, the impact of CO mass fraction distribution in terms of wind being directed from the southeast or simply south direction should also increase the pollutants in the less dense area. However, the distribution will vary based on the geometry, impacting the structural buildings and dwellers near the southeast zone. Figure 14 overall represents the simulated results of such circumstances. It could be observed that the distribution of pollutants across the single regular street is uniform due to the less hindrance in the mid-section (Figure 14a). However, during the existence of the single irregular road, the local maxima of CO mass fraction increased from 0.013 to 0.071, which is a significant increase of 446.15%, impacting the structures near the southeast zone, as seen in Figure 14b. Therefore, the increasing value of mass fraction of pollutants will serve as a warning to the public health. However, as double regular and irregular streets and multiple irregular streets were considered in Figure 14c–e, the CO mass fraction distribution downgraded, and the local maxima have been obtained to be 0.012, which is closer to that of a single regular street. This could be explained in light of the wind direction, which was coming from the southeast, and the buildings were serving as obstacles. Even the traffic emission were stuck across different structures, leading to lower concentration in between the roads.

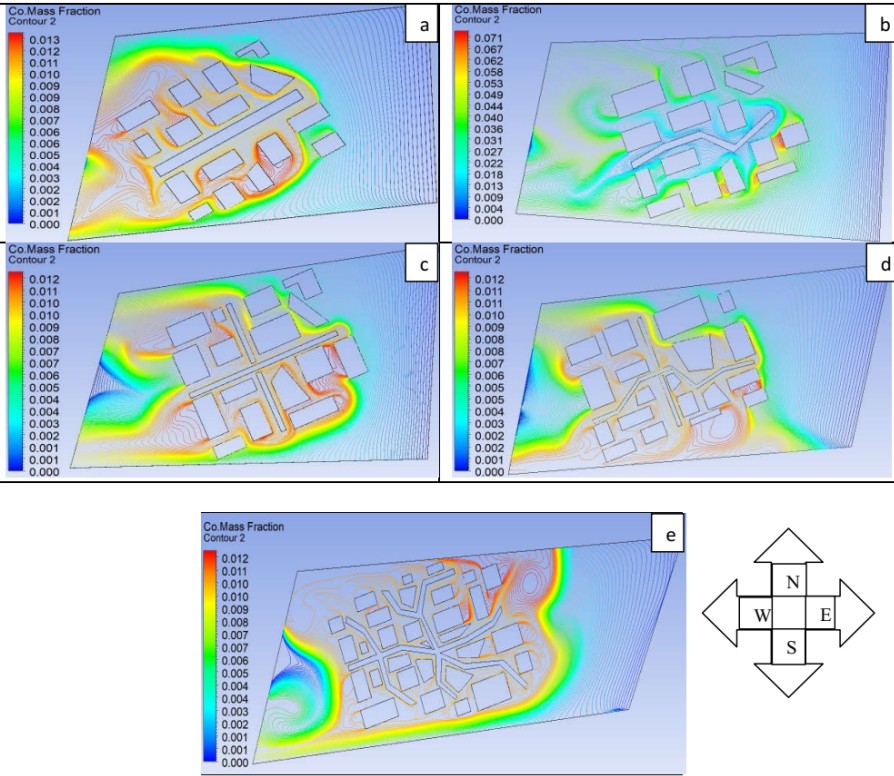

**Figure 14.** Concentration of traffic-exhausted CO mass fraction in terms from southeast wind flow of five different streets sequentially: (**a**) single regular, (**b**) single irregular, (**c**) double regular, (**d**) double irregular, and (**e**) multiple irregular.

### 4.3. Effect of Street Dimensions on Turbulent Kinetic Energy

The formation of vortices and turbulence has been observed during the case studies and wind impact. The turbulent kinetic energy (TKE) is one of the key attributes to define the intensity of the turbulence. The determination of TKE has already been presented in Equation (9). The formation of TKE needs to be maintained in order to maintain turbulence. Figure 15 illustrates the variations in TKE in terms of geometric variation of streets. In general, it could be observed that the concentration TKE generation as well as local maxima are found across the space mostly free from structural buildings or roads. However, like the CO mass fraction, the distribution of TKE will also be restricted outside the inhabitants' zones if irregular streets are present.

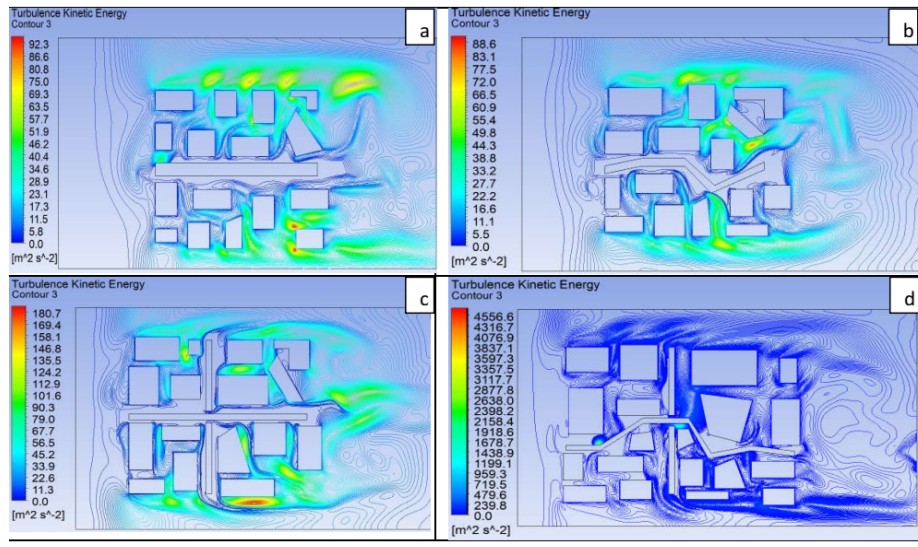

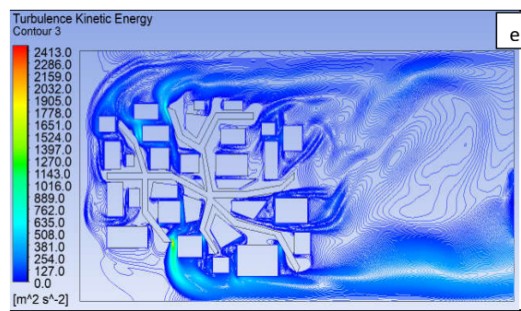

**Figure 15.** Variations in turbulent kinetic energy in terms of five different streets: namely, (**a**) single regular, (**b**) single irregular, (**c**) double regular, (**d**) double irregular, and (**e**) multiple irregular.

As per Figure 15a,b, the TKE was quantitatively 4.29% lower, as a single irregular street is considered in the geometry instead of a regular one. While the highest TKE has been recorded 92.3 $m^2/s^2$ for a single regular street, the value fell to 88.5 $m^2/s^2$. This could be explained in terms of atmospheric turbulence, which would be comparatively higher for a single regular street in the middle of the geometry as well as closer to the inlet. Therefore, the formation of vortices will be higher as the wind crossed the outlet. Meanwhile, due to the existence of an irregular single street, the air would keep crossing through obstacles at a lower velocity and will create small re-circulations. Therefore, TKE will be lower due to the lower intensity of the turbulence, as seen in Figure 15b. On the other hand, peak values of TKE significantly increased in terms of double regular and irregular streets as described in Figure 15c,d, respectively, with double irregular streets (Figure 15e) recording the maximum TKE amongst all streets. The maximum TKE value has been obtained to be 180.7 $m^2/s^2$ for a double regular street (Figure 15c), which increased to a massive 4556.6 $m^2/s^2$ for double irregular streets, as shown in Figure 15d. The sharp increase could be explained in terms of the increasing turbulence created due to the complexity of the street dimensions in the double irregular streets. The formation of vortices surrounding the buildings accumulated to the increasing peak values of TKE. Interestingly, the peak TKE is found to be 47.04% less in terms of multiple irregular streets, as shown in Figure 15e. While the double irregular streets create more space for increased turbulence, the multi-irregular streets and increased buildings did not provide adequate space for increasing vortices, and as a result, the peak TKE plummeted. However, as the air approached the outlet, the bigger formation of re-circulations was observed, which did not affect the inhabitants or industrial areas simultaneously.

## 5. Discussion of the Results

### *5.1. Purpose of Five Different Street Geometries and Major Findings*

#### 5.1.1. Single Regular Street

The single regular street is one of the five streets considered in this study to observe the influence of the geometry on the pollutants dispersion. It has been observed that the inhabitants dwelling near the single regular street are exposed to pollutants (CO mass fraction in this study) more if buildings have free space near the wind side. Therefore, the air has free space due to less obstacles, and it would not be a positive approach to build residential properties near the single regular street. Furthermore, the impact of a single regular street is monitored during the seasonal cycle as well. It has been observed that the northwest wind direction causes a CO mass fraction to spread throughout the neighboring buildings of the single regular street, and therefore, the negative impact remains the same for the people living in close proximity to the single regular street in the winter. The structural buildings away from the single regular street were exposed to a smaller concentration of pollutants. Meanwhile, considering the wind flowing from the south or southeast direction in the summer, the inhabitants near the southeast zone of the single regular street were more exposed to the pollutants compared to the inhabitants of the north region. Therefore, this would serve as evidence that inhabitants in the southeast region of Tejgaon and Gazipur are the worst victims of the pollutants.

#### 5.1.2. Single Irregular Street

The single irregular street is defined as a single road with a bend shape. Due to the increased hindrance and extra corners within the geometry, the wind velocity decreased in the mid-section surrounding the crossroads; however, the CO mass fraction distribution is around 8.33% higher than that of a single regular street. Under this set-up, the buildings and inhabitants close to the streets, inlet and outlet are the worst sufferers, as the bend shape of the single irregular street caused more blockage to the wind as well as pollutants. However, inhabitants facing toward the street had less exposure to the pollutants as air flows with an improved mobility over the street rather than through the canyons. It should be mentioned here that during the winter, the structural buildings near the southeast region are exposed to at least 25% more traffic emission than those of the northeast region despite the wind being directed from the northern zone. This is due to the fact that due to the existence of the single irregular street and increased corners, the wind faced certain obstacles and created small re-circulations, and some fractions of the pollutants are able to bypass the way toward the outlet due to more space in the southern zone. As a consequence, inhabitants near the far southeast zone are exposed to CO mass fractions more. In addition, as the wind flew from the southeast region directly, the situation exacerbated for the dwellers in the southern region as the concentration of the pollutants increased further, particularly at the neighborhood of the outlet. The CO mass fraction 0.071 suggest that the inhabitants faced 491.66% more pollutants over the summer season. So, the consideration of street canyons in Tejgaon and Gazipur areas should be skipped near the south or southeast region.

#### 5.1.3. Double Regular Street

A double regular street is considered as a crossroad of two single regular streets. This is one of the most common street types in Dhaka city due to the rapid surge in urbanization planning. Due to the presence of a crossroad, the concentration of pollutants was found to be quantitatively higher at the end of each street of the crossroad due to the absence of any structural building or obstacle. Therefore, the inhabitants living near the end of each crossroad experienced more pollutants, at least 30% higher. Therefore, it is recommended to create obstacles or plan trees near the end of intersection to ensure public safety. However, the obstacle should not block the free roaming of traffic, as it could lead to a road accident, which is another global problem. However, the situation in the winter downgraded further for the inhabitants near the double regular street, as the wind flows at lower speed compared to that in the summer season as well as in the rainy season. The

results presented in this study depict a similar accumulation of the pollutants inside the canyon in any case of the wind direction. In the winter season, due to the blowing wind in the northwest direction in Dhaka city, a higher amount of pollutants gather in the southeast region of the city.

### 5.1.4. Double Irregular Street

The term double irregular street is defined as a crossroad of a single regular and a single irregular street. The findings of this study suggest that the implications of a double irregular street reasonably mitigated the pollutants concentration by at least 61.51% due to increased blockage and corners within the mid-section, creating more opportunities for the pollutants to disperse through the free space where inhabitants will be less exposed. However, the exposure rate will increase under this set-up if inhabitants live near the east region during the winter as the wind flows from the northeast region and goes toward the east, which is also the outlet region. Therefore, the number of structural buildings should be limited near the east region to avoid creating threat to public health and safety. The similar observation has also been recorded during the summer season as well. Although the concentration remained similar to that of winter, it will remain a concern. Overall, the consideration of a double irregular street with a bit more planning will increase the living standard of the inhabitants.

### 5.1.5. Multi-Irregular Street

Multi-irregular streets are mostly common in Dhaka city near the industrial areas, particularly Tejgaon and Gazipur. The term is self-explanatory, and the representation of such remains closer to the real-life scenario in this study. Multi-irregular streets will contain the most obstacles for the wind, and therefore, the peak velocities under different parametric study are always found to be the highest. However, the geometry actually improved the pollutants' exposure rate to the inhabitants as due to the obstacles, the pollutants are dispersed through the free space from the north and south regions. Nevertheless, some of the regions near the east were exposed to a heavy concentration of pollutants, just like the double streets. These phenomena have also been observed in terms of the seasonal cycle, where buildings near the east or outlet were exposed to the highest CO mass fraction. Therefore, the east region requires more free space to increase public health safety. The wind typically blows from the southeast direction and the northwest region during the summer season, which is public heath friendly, as illustrated in Figure 14e. On the other hand, the wind flows from the northwest direction with low speed in the winter season compared to that of any season in the year. As a consequence, increasing concentrations of pollutants were observed in the southeast region in the city, as shown in Figure 13e.

### 5.2. Consideration of RANS over LES

The present study considered the RANS approach over the LES. Both options have been found to be accurate as reported by the literature. Blocken [48] reported that even though LES is considered to be superior to RANS to some extents, LES outcomes are more sensitive toward several user-defined computational settings and relevant variables, which may make each model more sophisticated. Furthermore, the RANS approach has been more practiced in the air pollutants dispersion in terms of street canyon and traffic emission. Therefore, the best practice benchmark has already been well-established by the researchers. However, the urban aerodynamics LES still has not been fully explored to an extent which can be considered as a benchmark solution. The guidelines on LES will depend on multifarious simulations as well as sensitivity analyses, which could be time-consuming. However, the RANS model is already a proven approach with a benchmark solution after the sensitivity analyses and has been in industrial practice for decades.

This study investigates the suburb-scale investigation for pollutants dispersion in two traffic-packed areas of Dhaka city. The purpose was to study the impact of different street geometries in a real-life scenario. The ideal approach would be to understand the impact

with proven evidence so that it can be further explored by the air quality investigators from the Government. Since the RANS model has been validated with the current geometries and variables, it provided more confidence on the present approach proceed. Therefore, LES is not separately explored considering the fact RANS is also an established method to study pollutants dispersion.

### 5.3. Ability of the Present Model as a Generic Model

Two of the major industrial areas of Dhaka—namely, Tejgaon and Gazipur—are considered as the areas of interest due to the suburbs filled with both residential and industrial buildings as well as heavy density of traffic. Five different geometries of street have been considered to investigate the wind velocity, influence of wind direction, and CO mass fraction. While the present study considers the situation of Tejgaon and Gazipur, a similar model could be extended as a generic model. In any country, those five types of streets will exist, and if the wind direction and traffic mobility can be integrated precisely, the model can still be considered to investigate any area across the globe where one of the five streets is present in a similar set up.

The weather data could be collected from the Bureau of Meteorology of each country, and information on the traffic mobility could be gathered from the Department of Transport or relevant Government body. Therefore, a similar approach to that presented in this study could be corroborated into studying the influence of different parameters in other geographical locations. Recently, Lin et al. [61] presented a virtual geographic environments (VGE) platform based on different environmental parameters. This indicates that prior research is a mandatory task before establishing such a framework.

### 5.4. Limitations of the Study and Future Plan

The present study considered traffic emission and CO mass fraction as a representative. However, the traffic-induced turbulence treatments and the chemical transformations are not considered to simplify the study. Furthermore, the investigation did not allow the effect of non-grid obstacles to the flow of air such as trees. Nevertheless, based on a real-life scenario, the number of trees is unfortunately quite limited due to the rapid urbanization and the requirement to free up space for more buildings. However, the Government of Bangladesh is strongly encouraging the tree plantation program. Therefore, the consideration of trees will be under consideration in future research to make the model more realistic.

Furthermore, the emission rate of the CO that corresponds to the traffic's volume speed of 6.4 km/h is computed as 215.34 tonnes per day. The background of concentration of the CO has not been taken into account in the computation using the ANSYS Fluent turbulence RANS $k - \epsilon$ model. The values, therefore, corresponded only to the contribution of the emission originating from local traffic in Dhaka city streets. Subsequently, the total concentration has been evaluated by adding the measurement of the averaged urban background concentration to the values computed by the CFD model. The current investigation considered the averaged urban background concentration of the CO pollutant. According to the traffic flow characteristic, in the less congested weekday traffic in Dhaka city street, approximately 698 vehicles move per hour [62,63]. The emission rate of the CO is computed based on the rational emission factors of vehicular traffic; these factors depend on the vehicular speed with various traffic fleet categories.

The existing Geographic Information System (GIS) teams of countries collect, organize, and analyze pollutants dispersion data. However, in order to let the GIS team work efficiently, the numerical model in different systematic geometries needs to be planned and correlated to understand the effect of important variables into the air pollutants dispersion. A similar approach has been taken into account by Badach et al. [64] while presenting their urban blue–green infrastructure tool for air quality management. Therefore, more investigations of different circumstances are required. In most of the developed countries, the use of unmanned aerial vehicles (UAVs) is gaining more interest in air quality

monitoring [65], and this could be one of the future recommendations for a developing country such as Bangladesh.

The rise of electric vehicles will certainly ensure a sustainable environment, but the electrical vehicle is not a sole solution, as the the maintenance cost of e-vehicles does not come cheap. Considering the economy of Bangladesh, it may take longer than anticipated to attain a sustainable framework or green energy. Therefore, the present situation of environmental pollution needs to be addressed by significant research and scientific attention. Bangladesh launched the "Bangladesh Vision 2041" where citizens of the country could monitor real-time air quality among other different attributes. The investigation presented in this study will extensively benefit the country and assist toward the fulfillment of such impressive goals. Our future research will investigate at a further meticulous scale including the consideration of trees into the design. The consideration of extracting the data to analyze through a machine learning algorithm is also under strong consideration of the authors.

## 6. Conclusions

The present study investigates the influence of street canyons and traffic emission (CO mass fraction) under the consideration of wind velocity and direction as well as seasonal cycle in two of the busiest areas of Dhaka: namely, Tejgaon and Gazipur. The distribution pollutants under different geometries of the streets have been investigated, and discussions have been added, keeping the sustainability framework of Bangladesh. The key findings of this study are the following:

- A single regular street creates fewer obstacles to the wind, and therefore, the inhabitants residing in close proximity are exposed to the highest concentration of pollutants from the traffic. The situation remains the same in the winter and summer seasons as well where dwellers near the single regular street experience approximately 50% and 53.84% more concentration of traffic emission compared to other inhabitants living reasonably far from the street. Therefore, more structural obstacles or trees would be required to mitigate the pollutants dispersion as well as a disciplined traffic system.
- A single irregular street reduces the exposure rate for the inhabitants near the street due to the existence of corners and obstacles by around 20%; however, the inhabitants near the east region still experience the most air pollutants. This suffering augments increasingly in the summer season when the wind blows from the southeast region. The density of traffic-generated pollutants increases by 491.66%, which is an alarming finding.
- A double regular streets canyon works well considerably to disperse pollutants. However, due to low wind velocity as well as much traffic in the cross-section region, the pollutants concentration was almost 10% higher than that of other parts of the canyon.
- Double irregular streets considerably reduced the pollutants concentration by approximately 8.33% in the mid-section due to the increased hindrance. However, the inhabitants near the east region still remain the worst victims with at least 10% more exposure to the CO mass fraction compared to that of inhabitants in other regions. This could also be linked with the fact that double irregular streets exhibited the most turbulence kinetic energy, which was at least 47.07% more than that of the multiple irregular streets.
- Multi-irregular streets behave as traps of pollutants due to the frequent changing of the wind-flowing directions. The inhabitants near the irregular streets are in more danger—by about 11.25%—compared to that of the far inhabitants.
- Multi-irregular streets create the most obstacles for the wind, leading to approximately 30% more vorticities in the free space compared to a double irregular street.

**Author Contributions:** Conceptualization, M.N.A., M.E.A., M.F.H. and M.M.M.; methodology, M.E.A., M.N.A., M.F.H. and M.M.M.; software, M.N.A., M.E.A. and M.M.M.; validation, M.E.A., M.N.A., M.F.H. and M.M.M.; formal analysis, M.E.A., M.N.A., M.F.H. and M.M.M.; investigation, M.E.A., M.N.A., M.F.H., M.M.M. and S.S.; resources, M.E.A., M.N.A., M.M.M. and S.S.; data creation, M.E.A., M.N.A., M.F.H. and M.M.M.; writing—original draft preparation, M.E.A., M.N.A., M.F.H. and M.M.M.; writing—review and editing, M.F.H., M.N.A., M.M.M. and S.S.; visualization, M.E.A., M.N.A., M.F.H. and M.M.M.; supervision, M.N.A. and M.M.M.; project administration, M.N.A.; funding acquisition, M.N.A. All authors have read and agreed to the published version of the manuscript.

**Funding:** This research has been funded by the Ministry of Science and Technology (MOST), Government of the People's Republic of Bangladesh during the fiscal year 2017–2018.

**Institutional Review Board Statement:** Not applicable.

**Informed Consent Statement:** Not applicable.

**Data Availability Statement:** The data that support the findings of this study are available from the corresponding author upon reasonable results.

**Acknowledgments:** All the authors gratefully acknowledge the corresponding author Most.Nasrin Akhter and the Dhaka University of Engineering and Technology (DUET) for providing the facilities needed to use ANSYS Fluent.

**Conflicts of Interest:** The authors declare no conflict of interest. The finders had no role in the design of the study; in the collection, analyses, or interpretation of data; in the writing of the manuscript; or in the decision to publish the results.

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
