# Peer review of "FVM-RANS Modeling of Air Pollutants Dispersion and Traffic Emission in Dhaka City on a Suburb Scale"

_sustainability, doi:10.3390/su15010673_

Round 1

Reviewer 1 Report

The study, ‘FVM-RANS modelling of air pollutants dispersion and traffic emission in Dhaka city on a suburb-scale’ is a very interesting work and nicely written. Also it is bringing some interesting fact which will be beneficial to the readers. However, I have some clarifications/suggestions which I would like clear before accepting this manuscript for publication.

English correction is required. Some grammatical and symbol errors are noticebale. Please correct the same.

I recommend to maintain the same directional orientation in all the figures. Fig 11 and 12 has North in the top whereas, fig 13 and 14 has East in the top. This can be avoided and this will help in better comparison of the figures.

The wind flow directions of south-east and north-east directions are analysed. Similarly, are the other directional flows are analysed. If not why?

Author Response

Kind Response to the Reviewer’s Comments

Title: FVM-RANS modeling of air pollutants dispersion and traffic emission in Dhaka city on a suburb-scale

Manuscript ID: sustainability-2075974

We take this opportunity to thank the editor and the reviewers of our paper for their kind collaborations to the improvement of this manuscript. We have taken into account all the concerns raised, and we have made the suggested modifications. We have implemented numerous improvements to the paper. Below we have justified our replies to the suggestions made by the respected reviewers of the paper.

Reviewer 2 Report

Review: FVM-RANS modelling of air pollutants dispersion and traffic emission in Dhaka city on a suburb-scale

This paper predicted the movement of pollutants due to traffic congestion by applying FVM as a RANS approach using CFD to carbon monoxide, a representative material of traffic congestion.

The results of reviewing this paper are as follows.

In the introduction, a suitable example of air pollution caused by traffic was given and clear information was provided on the city being measured.

In mathematical formulas, the equations that are clearly applied were explained.

In the methodology, the measurement location, wind direction, area, and model variables were clearly explained.

Model validation was clarified.

The expression of the results was clearly organized by dividing the case study into three categories, and the results were well organized in terms of wind speed, wind direction, and CO mass fraction.

The discussion of the results was also relatively clear.

In conclusion, this study is considered sufficient to be published in a sustainability journal by modeling the air pollutant movement in detail due to the expected traffic congestion in Dhaka.

Author Response

(The authors gave the same response as above.)
